# From layer-wise pruning to layer-over global rank optimization for adaptive sparse and low rank based deep learning model compression

**Donggyu Beom**[1] iD                                        BEOM9977@JNU.AC.KR

[1] *Department of Intelligent Electronics and Computer Engineering*
*Chonnam National University, Gwangju, Republic of Korea*

**Minyeong Park**[2]                                          QKRALSDUD25@JNU.AC.KR

[2] *Department of Computer Engineering*
*Chonnam National University, Gwangju, Republic of Korea*

**Suhyung Park**[1,2]                                         SUHYUNG@JNU.AC.KR

## Abstract

Deep learning has achieved substantial progress in biomedical image analysis, yet high computational and memory demands hinder deployment in resource-constrained clinical settings. While low-rank factorization via SVD is a prevalent compression strategy, conventional rank truncation introduces information loss that degrades performance under aggressive compression rates. In this work, we propose a layer-over rank optimization framework that decomposes layer-wise weight matrices into low-rank and sparse components while enabling adaptive rank allocation across layers under a unified parameter budget. Layer-wise optimization proceeds with principal component analysis (RPCA) that alternates between full-rank reconstruction and structured constraint enforcement by balancing reconstruction and task-specific losses to compensate for truncation-induced information loss. Additionally, global expansion of rank optimization controls layer-over flexible rank allocation, enabling dynamic distribution of representational capacity across layers. Extensive experiments on biomedical image analysis benchmarks confirm that the proposed method achieves superior accuracy and robustness over conventional low-rank compression methods, particularly under high compression rates.

**Keywords:** Model compression, low-rank, sparsity, layer-wise, layer-over.

## 1. Introduction

Deep neural networks (DNNs) have achieved strong performance in biomedical image analysis, yet their deployment remains challenging in resource-constrained environments due to high computational and memory demands. Low-rank factorization, particularly singular value decomposition (SVD), is widely used for model compression. However, conventional SVD-based methods rely on fixed rank truncation, which introduces information loss and degrades performance under aggressive compression. In addition, layer-wise rank selection ignores layer-over interactions, leading to suboptimal allocation of representational capacity across the network. To address this, we employ 1) low-rank and sparse decomposition framework of weight matrices for layer-wise optimization, and 2) a global rank allocation strategy across layers under a certain parameter budget within a single optimization framework. To this end, we refine low-rank approximation and sparse pruning structures by balancing full-rank reconstruction and task-specific losses under the framework of alternating direction method (ADM) to recover information loss during compression.

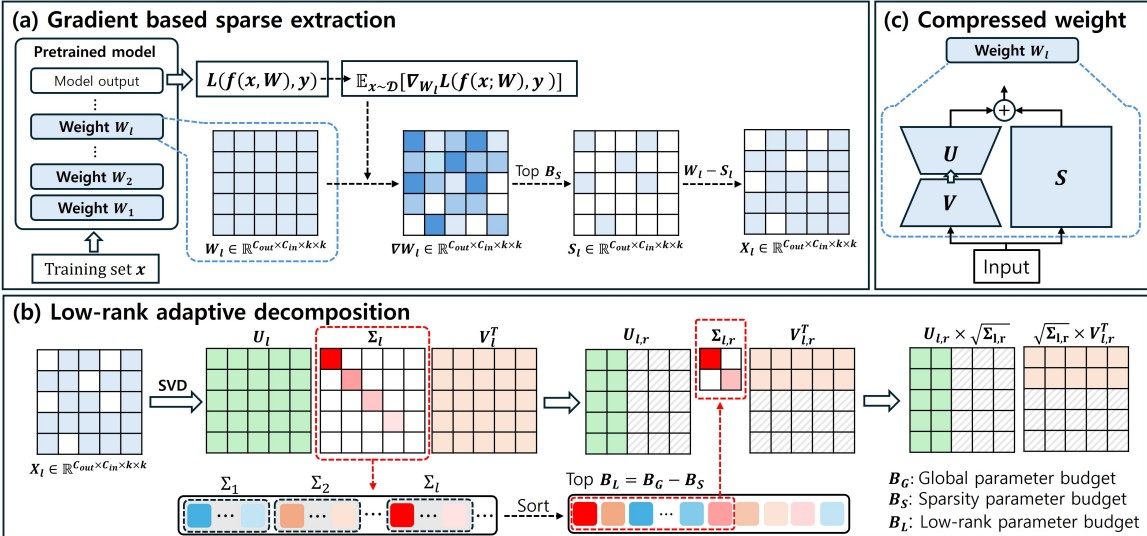

Figure 1: Overview of the compression process. (a) The sparse component is constructed by selecting parameters with large gradient magnitude over the training data. (b) The remaining weights are decomposed via SVD, and singular values are globally ranked across layers to allocate the low-rank budget. (c) The compressed weights are represented as a combination of low-rank and sparse components, $UV + S$, under a global parameter budget.

## 2. Methods and Results

**Layer-wise sparse and layer-over low-rank optimization:** We decompose pretrained weights into low-rank and sparse components, where sparse weights are selected based on gradient magnitude and the remaining weights are approximated via rank factorization. The singular values are then globally ranked across layers and selected considering the parameter budget. This enables more aggressive compression of inherently low-rank layers while preserving important global structures. (Fig.1).

**Local and global unified optimization framework:** To mitigate truncation-induced information loss, we cast a mathematical framework as:

$$\min_{X,U,V,S} \frac{1}{2}\|X - UV - S\|_F^2 + \lambda_s \|S\|_1 + \mathcal{R}(X,U,V,S) \quad \text{s.t.} \sum_{l=1}^{L} r_l \le B_L \qquad (1)$$

This formulation incorporates a task-aware regularization term $\mathcal{R}(\cdot)$ and is optimized via an ADM-based iterative scheme that alternates between full-rank reconstruction and structured updates of $U$, $V$, and $S$, enabling recovery of task-relevant information beyond a fixed low-rank subspace. Here, $r_l$ denotes the rank of layer $l$, and $B_L$ is the global low-rank parameter budget that controls the total rank allocation across all layers.

**Dataset and implementation.** We evaluate our method on the Breast Ultrasound Image(BUSI) dataset(Al-Dhabyani et al., 2020) with images resized to $256 \times 256$ following UNeXt(Valanarasu and Patel, 2022). UNeXt is used as the backbone with BCE and Dice loss and the Adam optimizer.Compression is performed using the proposed optimization framework, where each stage is optimized for 100 epochs and repeated for 100 iterations. Experiments are conducted on a single NVIDIA RTX 3090 GPU.

**Results.** With varying low-rank parameter budgets (Table 1, top), with sparsity fixed at 10%, stable performance is maintained up to 20% compression. In Table 1 (middle), sparsity is varied from 5% to 25% under a fixed global parameter budget (50%), with optimal performance at 10%. In Table 1 (bottom), UNeXt shows comparable performance to UNet(Ronneberger et al., 2015). Under compression, existing methods(FWSVD(Hsu et al., 2022), ASVD(Yuan et al., 2025)) relying on low-rank approximation show limited gains, while the proposed method achieves greater improvement using 40% low-rank with 10% sparsity. Across all settings, compressed models outperform the pretrained model, suggesting that redundant weights are effectively removed through low-rank decomposition while global rank allocation redistributes representational capacity across layers to preserve important structures. Furthermore, the combination of low-rank and sparse modeling provides a more compact yet expressive representation, improving both efficiency and robustness.

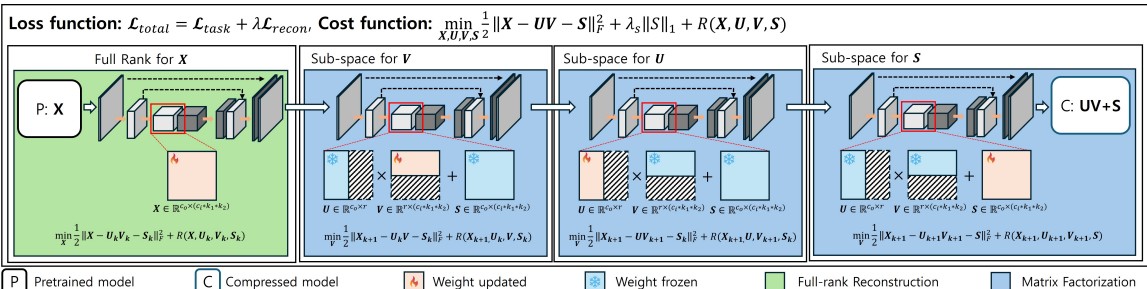

Figure 2: Overview of the proposed method. The top diagram illustrates the global rank optimization framework across layers, where each stage alternates between full-rank reconstruction and subspace updates. The bottom diagrams detail the updates within a single stage, including full-rank reconstruction for $X$ and subspace optimization for $V$, $U$, and $S$, followed by weight reconstruction via $UV + S$.

| Performance with varying low-rank budgets (with sparsity rate of 10%) | | | | | | |
|---|---|---|---|---|---|---|
| | UNeXt | 50 | 40 | 30 | 20 | 10 |
| mIoU | 0.6367 | 0.6999 | **0.7006** | 0.6923 | 0.6902 | 0.6765 |
| Dice | 0.7769 | 0.8224 | **0.8229** | 0.8169 | 0.8157 | 0.8040 |

| Performance with varying sparsity budgets (with total compression rate of 50%) | | | | | | |
|---|---|---|---|---|---|---|
| | UNeXt | 5 | 10 | 15 | 20 | 25 |
| mIoU | 0.6367 | 0.6785 | **0.6934** | 0.6793 | 0.6743 | 0.6881 |
| Dice | 0.7769 | 0.8066 | **0.8170** | 0.8081 | 0.8039 | 0.8143 |

| Comparison with existing methods under the same parameter budget | | | | | |
|---|---|---|---|---|---|
| | UNet | UNeXt | FWSVD(50%) | ASVD(50%) | Proposed(40%) |
| mIoU | 0.6451 | 0.6367 | 0.6456 | 0.6387 | **0.7006** |
| Dice | 0.7807 | 0.7769 | 0.7826 | 0.7770 | **0.8229** |

Table 1: Quantitative evaluation under different compression settings. The top section shows results across varying low-rank parameter budgets with a fixed sparsity parameter budget (10%). The middle section presents results under different sparsity parameter budgets with a fixed global parameter budget (50%). The bottom section compares the proposed method with existing approaches under the same budget, where baselines use a 50% low-rank budget and the proposed method uses 40% low-rank with 10% sparsity.

## Acknowledgements

This work was partly supported by 1) the National Research Foundation of Korea (NRF) grant funded by the Korean government (MSIT) (RS-2024-00357917, 34%), 2) the Institute of Information & Communications Technology Planning & Evaluation(IITP)-Innovative Human Resource Development for Local Intellectualization program grant funded by the Korea government(MSIT)(IITP-2026-RS-2022-00156287, 33%), and 3) Basic Science Research Program through the National Research Foundation of Korea(NRF) funded by the Ministry of Education (RS-2025-25398164, 33%)

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
