# OpenReview forum: "From layer-wise pruning to layer-over global rank optimization for adaptive sparse and low rank based deep learning model compression"
_MIDL.io/2026/Short_Papers — MIDL 2026 - Short Papers Poster_

### Official Review · Reviewer_PJ4B · 2026-05-03
**Gradient-based parameter importance selection for SVD weight compression**

**Rating:** 4
**Confidence:** 3

**Review:**

Please see Strengths and Weaknesses below.

**Summary:**

The authors propose a technique for model compression that uses gradient magnitude-based weight sparsification with a subsequent application of SVD. They test on breast ultrasound segmentation via UNeXt. They compare to other SVD-based techniques and find decent improvements given the same parameter budget, indicating a potentially promising direction.

**Strengths:**

- The idea is reasonable and implemented suitably, and due to its general formulation, could be applied to any sort of model, beyond medical imaging ML, or even computer vision. If the authors wish to extend this to a more general ML publication, I suggest examining this similar EMNLP 2025 paper as related work: https://aclanthology.org/2025.emnlp-main.1338/, which I found via a quick chatbot query.
- Preliminary results are promising and sufficiently comprehensive for a MIDL short paper, and compared to reasonable SVD baselines. Shows notable performance improvements given the same parameter budget.
- Showing performance sensitivity to both low rank and sparsity budgets is a key experiment, and the results are helpful and show promise for the generality of the approach.

**Weaknesses:**

While it is challenging to fit this in a MIDL short paper, the method would benefit from comparison to related work to properly contextualize the contribution and its significance (see e.g. the EMNLP 2025 similar work I found as mentioned above after a quick search). It is a fairly natural combination of existing compression techniques (gradient-based parameter importance selection/sparsification and SVD), so I wouldn't be surprised if other similar works have been created.

**Justification Of Rating:**

The authors' proposed model compression technique shows promising preliminary results on a medical image segmentation task. I'm curious how it compared to other similar related work in the broader machine learning space, as well as how it might extend to other domains beyond medical image analysis and computer vision.

---

### Decision · Program_Chairs · 2026-05-08

Accept (Poster)